# Building a public health workforce for a university campus during a pandemic using a practicum framework: Design and outcomes

**Carolyn S. Dewa** [1,2]*, **Zoe Che**[1], **Andrea M. Guggenbickler**[1], **Rebecca Phan**[1], **Bradley Pollock**[1]

1 Department of Public Health Sciences, University of California Davis, Davis, California, United States of America, 2 Department of Psychiatry and Behavioral Sciences, University of California Davis, Davis, California, United States of America

* csdewa@ucdavis.edu

**Data Availability Statement:** The data relevant to this study are available from the University of Michigan openICPSR project site at https://doi.org/10.3886/E172841V1.

## Abstract

### Background

The reopening of college and university campuses was seen as presenting a high risk for transmission of COVID-19. Thus, these institutions faced with a new public health challenge never heretofore faced on this scale. To magnify the problem, they needed to rapidly develop and implement re-opening plans in an environment filled with uncertainty and for a population that was significantly less likely to observe COVID-19 mitigation behaviors. In response, within three weeks of opening, as part of its COVID-19 public health strategy, a West Coast university created and trained a public health workforce comprised of 282 undergraduates tasked with encouraging compliance with COVID-19 mitigating healthy behaviors.

### Main objectives

This paper describes the use and outcomes of a practicum framework to quickly create a university-based public health workforce. It addresses two questions: (1) Using a practicum framework, what are important considerations in designing and building a public health workforce for a university campus? and (2) What are the benefits to the workforce in terms of public health education and professional growth?

### Methods

Program administrative data were used to describe the workforce and their learning outcomes.

### Results

The majority of students indicated that through the practicum, they learned new skills/developed new attitudes (71.7%) and became aware of their own strengths and opportunities for professional growth (73.7%). The types of new skills and attitudes learned included communication (49.2%), conflict management (20.4%), time management (7.5%), and open-

**Funding:** The author(s) received no specific funding for this work.

**Competing interests:** The authors have declared that no competing interests exist.

mindedness/less judgmental attitude (14.6%). In terms of public health, they gained an understanding of infectious disease prevention (40.9%) that is multi-disciplinary (20.5%), and involves a community effort (36.8%).

## Conclusions

These findings demonstrate an effective way of rapidly addressing public health concerns that allowed for on the job training and opportunities for young adults to learn and grow. The practicum framework allowed the expeditious development of a public health workforce that ensured a fit between student interests and the role. This led to high retention with the majority of students continuing into the winter quarter. Only 5% of students reported not being satisfied with their position. None of the students contracted job-related COVID-19. The role gave students a sense of purpose during the pandemic's uncertain times that helped to protect them from the negative effects of stress. The practicum structure and support fostered a safe environment in which students were able to feel part of the larger community while gaining valuable work experience and skills and serve their community.

## Introduction

Young adults between 18 and 29 years old comprise 23% of all COVID-19 cases in the US [1]. In addition, there was evidence that young adults between 18–29 years of age are significantly less likely to observe COVID-19 mitigation behaviors such as mask wearing, physical distancing, and hand washing [2]. With this, the reopening of college and university campuses was identified as presenting a high risk for transmission of SARS-CoV-2 [2, 3]. Indeed, a number of high profile universities opened only to be forced to shut down a short time later because of campus COVID-19 outbreaks [4].

However, it has been argued that it is possible to limit the spread of COVID-19 on campuses through a combination of mitigation strategies [5]. Ensuring compliance with preventative behaviors such as masking, physical distancing, and hand washing, has been identified as one of the key strategies [5–7]. In addition, the US Centers for Disease Control and Prevention (CDC) recommends that reopening plans for institutions of higher education include strategies that help to enforce compliance with recommended COVID-19 mitigating behaviors such as provision of adequate supplies (e.g., face coverings, hand sanitizers) and messaging that is simple, clear, and accessible [7].

Van den Brouche [8] suggests that health promotion approaches are also key to encouraging adoption of preventative behaviors. This includes helping people understand that COVID-19 mitigating behaviors are effective in decreasing infection risk and that they are capable of doing them. In addition, effectively promoting COVID-19 mitigating behaviors involves changing social norms (i.e., wearing masks and physical distancing) and working towards a common good [9]. Furthermore, these health promotion messages must come from credible sources [9]. For example, peer educators have been shown to be effective messengers [10].

Peer education, though relatively underappreciated in higher education, finds its effectiveness through informal learning, establishing common ground, and recognizing unique perspectives, especially when utilized in social and health education [11]. Thus, peer education can be one of the ways to introduce COVID-19 policies to university campuses.

But, to provide effective education, peers must be prepared with knowledge and communication skills. In public health, practicum structures are used as a teaching vehicle. Practicums present students opportunities to apply public health knowledge and skills in practice [12]. Undergraduate practicum projects have been shown to be highly effective to train students for public health work [13]. Thus, the practicum is considered a capstone in an undergraduate student's public health experience [14]. This assumes that the practicum is being offered within the context of a public health major, raising the question of the effectiveness of a practicum framework to introduce and teach public health to students who are not majoring in public health. Furthermore, there is a paucity of literature on the use of the practicum model on a large scale to address a community's public health needs that a pandemic demands. These demands include continuously keeping up to date with quickly changing scientific knowledge, public health policies, and educating a potentially frustrated public while creating a positive learning experience.

Using a practicum framework, a West Coast university with an undergraduate enrollment of 31,162 students, developed a public health workforce of undergraduate peer educators to address its reopening plans and meet its COVID-19 health promotion mandate. In this paper, we describe the use and outcomes of a practicum framework to create a university-based public health workforce to limit the spread of COVID-19 on a university campus. We address two questions: (1) Using a practicum framework, what are important considerations in designing and building a workforce for a university campus? and (2) What are the benefits to the workforce in terms of public health education and professional growth? This paper contributes to filling the gap in the literature about how a practicum framework can be used on a large scale as an effective learning tool in an urgent public health crisis.

## Background

In August 2020, as part of its COVID-19 reopening plan, the university identified the need for a public health workforce. The novelty of the approach was that the workforce would be comprised of undergraduates who would be tasked with encouraging compliance with COVID-19 mitigating healthy behaviors. The program was launched as the Public Health Ambassador Program (PHAP). To optimally meet campus needs, it was determined that the workforce should be in place prior to the campus' reopening scheduled for the last week in September 2020 when it was anticipated that about 2,000 students would be moving into campus housing and another 18,000 would be returning to the area to live off-campus.

In addition to promoting public safety, the program was also envisioned as an opportunity to introduce interested students to public health sciences. More than a work program, it was structured as a practicum for which students were hired and allowed to enroll for one unit of public health sciences course credit. Thus, there was a strong emphasis on training and the educational aspect of the position.

## Methods

### Recruitment

Unlike typical courses, a three-step process was developed for students to apply for the limited number of PHAP positions. To review each application, interview applicants, and prepare them before the start of the school year, a staggered schedule was devised.

Step 1 began with recruitment. To attract diverse students, a position description was posted on the University job portal for undergraduate students seeking on- and off-campus jobs. The posting was also sent out to the campus community through the University President's weekly COVID-19 update as well as through academic department list serves. Interested

students were instructed to submit an application in the form of their resume and a cover letter.

## Application review

A multi-stage standardized review process was developed. In Stage 1, all applications were screened for: (1) completeness and (2) grammatical errors/typos in the cover letter or resume. Incomplete applications or those with more than four grammatical errors/typos did not proceed to Stage 2 of the review process. The decision to use this screen was based on the fact that to be effective, selected students would need to be detail oriented to ensure they absorbed the changing educational material.

At Stage 2, each application was reviewed by two independent reviewers from the PHAP graduate student supervisor team. Applications were reviewed and scored with respect to four criteria: (1) Team Player (e.g., participation in team sports, service groups, service on executive boards, musical bands), (2) Communication Skills (e.g., work experiences in customer service or tutoring), (3) Public Health Knowledge (e.g., experience with public health activities, public health minor), and (4) Experience with Healthy Behavior/Health Promotion Activities (e.g., work experience with a health promotion project). Each criterion was rated on a 3-point scale (0 = no evidence, 1 = some evidence, 2 = strong evidence). Reviewer scores were compared and discrepancies were discussed until consensus was achieved. Applications with total scores of one or more proceeded to Stage 3. Thus, this stage was used to evaluate a minimum requirement.

## Interview process

In the third step of the process, groups of 6–12 applicants were invited for virtual group interviews. Interviews lasted 1-hour. About 81.8% of the interviews were conducted with a committee of three interviewers (n = 297). The remaining 18.2% were conducted with a committee of two.

Using the Council on Education for Public Health (CEPH) guidelines for undergraduate public health education to identify important public health competencies [14], each applicant was independently scored by at least two interviewers on a 5-point scale on six dimensions: (1) attitude toward COVID-19 guidelines, (2) adaptability, (3) teamwork, (4) communication, (5) conflict management style, and (6) customer service. Table 1 contains the interview questions, the dimension assessed by each question, and the criteria for each question's scores.

Each interviewer's scores for the six dimensions were totaled and averaged. The cut-offs were used: (1) 6–14.9 points = *Decline*, (2) 15–23.9 points = *Maybe*, and (3) 24–30 points = *Accept*. About 87.1% interviewees fell into the *Maybe* range of scores (n = 316). Applicants whose scores fell into the *Maybe* category were discussed using a review of interview notes until consensus was reached. Discussions were held within 30 minutes of each interview session.

## Description of training process

All public health ambassador (PHAs) were required to undertake a total of 20 hours of asynchronous and synchronous education and training. All asynchronous training modules were accompanied with quizzes that emphasized the information to be mastered. Each PHA was given two attempts to pass each quiz with a score of 80%.

In addition to understanding of the epidemiology of COVID-19 and the university's policy for effective preventative behaviors, the curriculum covered CEPH recommendations of teamwork and leadership, networking and communication skills, fostering of community

**Table 1. Interview scoring criteria.**

| Dimension | Interview Question | Score Criteria | | | | |
|---|---|---|---|---|---|---|
| | | Score of 5 | Score of 4 | Score of 3 | Score of 2 | Score of 1 |
| **Attitude towards COVID-19 guidelines** | Please share your favorite hobby or pre-COVID activity. Tell us how you have adapted during COVID. | Student's response includes consideration for keeping others' safe or adherence to public health guidelines. | Student talks about COVID and how it has impacted them. | Student describes how they have adjusted during COVID without explaining why. Example: Student says they continued hobby inside. | Student does not go into detail or discuss specifics about adaptations. Student does not mention COVID. | Student may suggest they are no observing public health guidelines. |
| **Desire to help others** | What excites you most about this position? Why? | Student expresses sincere interest in public health and community advocacy. Student talks about an opportunity to grow and help others while making a change in their community. | Student expresses interest in public health or community betterment. Student talks about helping others and references an opportunity to feel like they are making a change. | Student discusses general interest in community betterment. Student discusses wanting to help others and campus. | Student discusses feeling ready to be back on campus as the main reason for applying. Student expresses some interest in public health or helping others. | Student talks about being bored or having nothing better to do. Student does not express interest in helping others or bettering the community. |
| **Adaptability** | This program will put you in unfamiliar situations. Please describe how you adapt to new and changing situations. | Student provides an example of how they adapted to a new situation and the steps they took. | Student cannot describe a specific example. Student discusses how they changed their behavior and whether they were successful. | Student talks about their ability to change behavior depending on the situation. | Student does not describe an example and does not discuss how they changed. | Student does not address the question. |
| **Teamwork** | Describe a time when you worked with someone with different beliefs. How did you deal with this situation? This could include team sports, a group project, etc. | Student provides specific example encountering a major difference in beliefs with someone they worked or interacted with. Student discusses specific steps they took to create a positive working relationship with the individual to accomplish a common goal. | Student discusses having a major difference with someone's work ethic, social beliefs, or COVID beliefs. Student talks about how they approached compromise depending on the common goal (working together, living together, etc.). | Student talks about a group project or instance, where they wanted to execute a project in a different way than someone else. Student discusses coming to a compromise. | Student talks about ignoring, reacting with anger, dismissing others because of differing beliefs. Student does not describe a solution or compromise to address the problem. | Student does not answer the question. |
| **Customer Service** | Student attitude and interaction with other students (e.g., professionalism respectfulness, friendliness, positive attitude) | Student is approachable and personable. Student is professional and respectful. Student interacts with other students with ease. | Student has a positive demeanor, has a background in customer service, is attentive when others are speaking, and is respectful of everyone. | Student pays attention to others when speaking, and/or is respectful during the interview. | Student is attentive but is quiet and not approachable. | Student is not attentive, arrives late, or is condescending towards others. Student is distracted (e.g., on the phone or distracts others. |
| **Communication skills and conflict management style when responding to a difficult situation** | You are a Public Health Ambassador and you notice someone on campus is not wearing a mask. You go up to them and kindly ask them to wear a mask. They argue with you. What would you do? | Student says they would deescalate the situation, and talk to the person about why they believe what they do and then try to tailor safe public health practices to that individual. Student shows compassion, understanding, and a positive attitude in this situation. | Student says that they would deescalate the situation and try to find out how they could help this individual in a calm manner. They would then try to educate them on campus policies and be respectful before potentially getting a supervisor. | Student discusses trying to be calm and to educate and ask them again to wear a mask. Student says if all else fails they will find a supervisor. | Student talks about immediately getting a supervisor or says they do not believe they can do anything in that situation to change that person's mind. | Student says they would immediately call the police or they say that they would not want to be in this situation. Student says they would argue back and believes aggression is the best response. |

Questions presented to each interviewee. Scores (1–5) provided by interviewers based on applicant answers. Includes each dimension that the interview questions were created to assess.

dynamics, and advocacy for and protection and promotion of public health [14]. Asynchronous module topics included: *COVID-19 University Guidelines*, *Introduction to the Epidemiology of COVID-19*, *Communication Skills*, *Public Health Ambassador Program Values*, and *Creating a Self-Care Plan*.

PHAs also underwent synchronous training via Zoom. The training provided an opportunity to practice communication skills based on likely scenarios. As suggested by CEPH, emphasis was given to developing students' critical thinking, professionalism, and advocacy skills for the health and well-being of the campus and community [14]. Ongoing coaching was provided by graduate student teaching assistants and health educators for in the moment learning opportunities.

## Self-reflection and learning

In the 10th week of quarter, PHAs were sent links to an online post-practicum course evaluation questionnaire. The protocol for the analyses of this dataset was reviewed by the university's Institutional Review Board and was deemed not to be human subject research. To evaluate the saliency of the practicum to student learning, students were asked to reflect on what they learned as PHAs, with three open-ended questions: (Question1) "What did your experiences as an PHA teach you about yourself this quarter?", (Question 2) "What was your best/favorite experience of being an PHA this quarter?", and (Question 3) "What did you learn about public health from being an PHA?" In addition, PHAs were asked to rate their role satisfaction with the questions, (Question 4) "Overall, I find my role as an PHA is:" on a 4-point scale (1 = Extremely Satisfying, 2 = Somewhat Satisfying, 3 = "Not Very Satisfying", 4 = "Not Satisfying"), (Question 5) "Please indicate the overall educational value of the APHA program" on a 5-point scale (1 = "Excellent," 2 = "Very Good," 3 = "Satisfactory," 4 = "Fair" and 5 = "Poor) and (Question 6) "The usefulness of the APHA program and training materials for my professional interests:" on a 5-point scale (1 = "Excellent," 2 = "Very Good," 3 = "Satisfactory," 4 = "Fair" and 5 = "Poor). The categories were aggregated into two categories, "satisfied" and "dissatisfied." Ambassadors' responses were compared to their sex and major to determine if there was a significant difference between their satisfaction.

Using thematic analysis, patterns and themes were identified and applied to the data; categories by main themes were developed for each question [15]. Responses to the open-ended questions were independently coded by RP and ZC. Codes for the first 30 responses for each of the questions were compared for the two analysts. There was 86.7% (26/30) agreement for Question 1, 100% (30/30) agreement for Question 2, and 83.3% (25/30) agreement for Question 3. Where there was disagreement, a third analyst (AG) was consulted and discussed as a group until consensus was reached. From this, a list of definitions for each code was created. The resulting codes were utilized to analyze the responses to the three open-ended questions (Table 2).

## Analysis

The frequencies of each of the three open-ended question response categories were calculated. In addition, for each interview question, scores were used to create two groups to indicate: (1) high interview score (score ≥4) and not high interview score (score < 4). Chi-square tests were used to test for a significant association between the interview groups (high score versus not high score) and the responses to the three open-ended questions. An independent t-test was used to test for a significant difference between the two interview groups and satisfaction with their PHA role.

**Table 2. Definitions of types of lessons learned.**

|  | Definition |
|---|---|
| **THINGS LEARNED** | |
| Communication Skills | Learned to converse with others more effectively and/or facilitating conflict management |
| Conflict Management | Learned to initiate conversations regarding noncompliance and/or how to de-escalate situations |
| Time Management | Learned to maintain work-life balance |
| Open-Minded | Learned to be less judgmental and/or more open-minded |
| Self-Realization about Strengths/Limitations | Learned anything personality traits, work style, self-worth, and/or self-confidence |
| Self-Worth | Recognized ability to make an impact |
| Self-Confidence | Become more confident in their skills, such as noncompliance |
| Patience | Recognized ability to be patient |
| **POSITIVE EXPERIENCES** | |
| Building Connections | Networked and/or fostered relationships |
| Building Relationships with a Diverse Group | Met people they would not have been able to if they were not involved as an ambassador |
| Community Building | Served the community and/or campus, bettered the community, and/or contributed to community health |
| Helping Others | Being helpful and/or helping community and/or campus |
| **ROLE OF PUBLIC HEALTH** | |
| Infectious Disease Prevention | Preventing COVID-19, COVID-19 guidelines and policies |
| Multidisciplinary | Public health as a multifaceted, team effort between multiple entities |
| Community Effort | Public health requiring community involvement to be successful |
| Challenges | Recognized that public health officials face barriers and obstacles |
| Essential | Recognized how essential and/or important public health is for communities |

Codes and definitions for responses to Questions 1, 2 and 3.

## Results

### Identifying the PHAs

In total, 607 students submitted applications for the PHA Program. In the first stage, 179 applications were screened out. In the second stage, 428 applicants were invited for interviews. Of these, 363 accepted the invitation (Fig 1).

Of those interviewed, 12% (n = 44) had total interview scores that fell within the *Accept* cutoff (total score between 24–30); one percent (n = 3) were at or below the *Decline* cut-off (total score less than 14.9). The remaining 87% (n = 316) scores fell within the *Maybe* category (total score between 15–23.9). Following discussion among the interviewers, 82% (n = 260) were *Accepted* and 18% (n = 56) were *Declined*. In the two weeks between interviews and the beginning of the school term, 2.0% (n = 5) of the students became unavailable for the Fall Quarter. Between the first and last week of the 10-week Fall Quarter, 5.6% (n = 17) of PHAs dropped out of the program leaving a total of 282 PHAs at Week 10.

PHAs represent a diversity of 56 undergraduate majors, with 5.9% of students majoring in the Humanities (n = 18), 4.3% from Engineering (n = 13), 60.9% from Lab Sciences (n = 185), 23.4% from Social Sciences (n = 71), 4.3% from Agricultural Sciences (n = 13), and 1.3% from Formal Sciences (n = 4). Among these, 10.5% (n = 32) were first years, 28.9% (n = 88) were second years, 36.2% (n = 110) were third years, 23.4% (n = 71) were fourth years (1%, n = 3 were missing years).

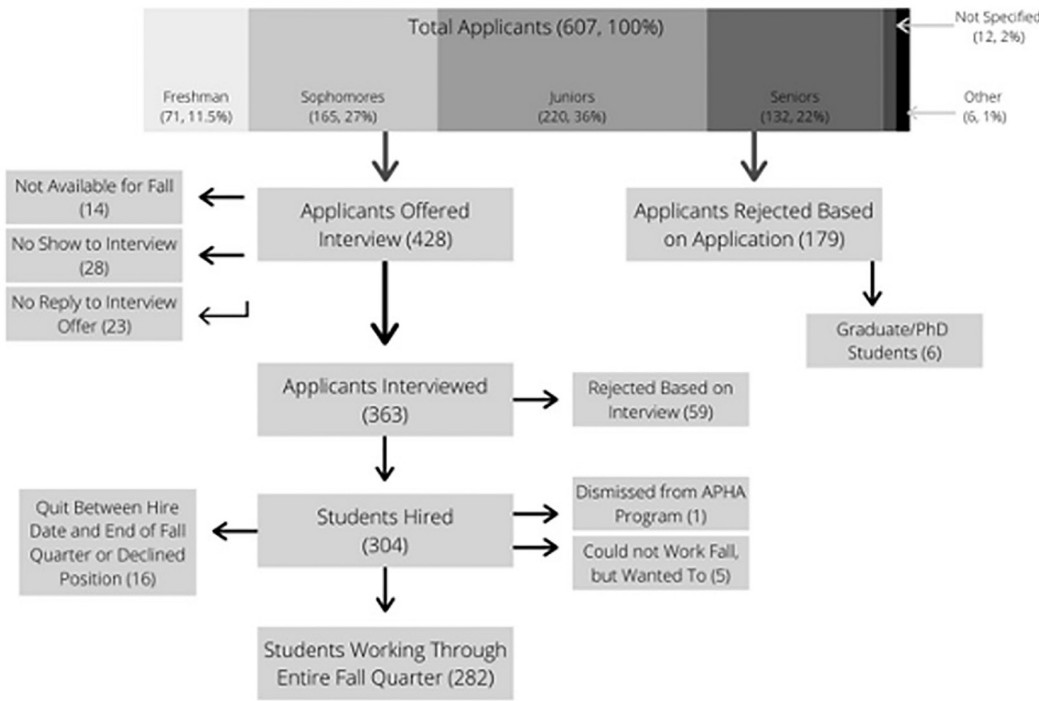

**Fig 1. Screening and selection process results.**

### Interview scores

The mean total interview score for the *Accepted* (n = 24) category was 24.7 (sd = 0.6). For the *Declined* (n = 3) category, the mean total score was 14.3 (sd = 0.59). The *Maybe* (n = 336) group had a mean total score of 20.8 (sd = 1.97). Those who were hired from the *Maybe* group had a mean total score of 20.6 (sd = 1.63). In contrast, those who were declined in the *Maybe* group with a mean score of 18.4 (sd = 1.2).

### Satisfaction with role

Of the 282 PHAs who continued with the program through the end of the Fall Quarter, 88.3% (n = 249) completed the final Fall quarter questionnaire. There were no significant differences between characteristics of those who did and did not complete the questionnaire with regard to their initial interview score (t-test (df = 280) = 1.15, p = 0.25) and year in school ($\chi^2$(df = 4) = 4.55, p = 0.34).

Of the PHA respondents, 98.8% rated their satisfaction with the PHA role (Table 3). The majority indicated they were satisfied with their role (95.2%, n = 238). Few were dissatisfied with their role either (4.8%, n = 12). In addition, 96.8% (n = 242) were satisfied the educational value of the program and 94.0% (n = 235) were satisfied with the value for their professional interests. There were no significant differences by either gender or STEM/Non-STEM major. Furthermore, none of the PHAs developed COVID-19 as a result of their role.

### What PHAs learned

Of the PHAs who completed the questionnaire, 96.4% (n = 240) responded to the question, "What did your experiences as an PHA teach you about yourself this quarter?" Table 4

**Table 3. Comparing gender and major to satisfaction.**

| | Role Satisfaction | | | | Satisfaction with Educational Value | | | | Satisfaction with Value for Professional Interests | | | |
|---|---|---|---|---|---|---|---|---|---|---|---|---|
| | Satisfied | | Dissatisfied | | Satisfied | | Dissatisfied | | Satisfied | | Dissatisfied | |
| | % | n | % | n | Test Statistic and p-value | % | n | % | n | Test Statistic and p-value | % | n | % | n | Test Statistic and p-value |
| Total | 95.2% | 238 | 4.8% | 12 | $X^2$(df = 1) = 204.3 p < 0.0001 | 96.8% | 242 | 3.2% | 8 | $X^2$(df = 1) = 219.0 p < 0.0001 | 94.0% | 235 | 6.0% | 15 | $X^2$(df = 1) = 193.6 p < 0.0001 |
| Gender | | | | | | | | | | | | | | | |
| Male | 19.3% | 46 | 33.3% | 4 | Fisher's Exact p = 0.27 | 19.4% | 47 | 27.5% | 3 | Fisher's Exact p = 0.2 | 19.6% | 46 | 26.7% | 4 | Fisher's Exact p = 0.51 |
| Female | 80.7% | 192 | 66.7% | 8 | | 80.6% | 195 | 62.5% | 5 | | 80.4% | 189 | 73.3% | 11 | |
| Major | | | | | | | | | | | | | | | |
| STEM | 87.0% | 207 | 100% | 12 | Fisher's Exact p = 0.37 | 87.6% | 212 | 87.5% | 7 | Fisher's Exact p = 1 | 88.1% | 207 | 80.0% | 12 | Fisher's Exact p = 0.41 |
| Non-STEM | 13.0% | 31 | 0.0% | 0 | | 12.4% | 30 | 12.5% | 1 | | 11.9% | 28 | 20.0% | 3 | |

Data from Questions 4, 5 and 6.

describes what respondents learned. They indicated they developed new skills/new attitudes (71.7%, n = 172) and became aware of strengths and opportunities for growth (73.7%, n = 175). The types of new skills and attitudes included communication skills (49.2%, n = 118), conflict management skills (20.4%, n = 49), time management skills (7.5%, n = 18), and open-mindedness/less judgmental attitude (14.6%, n = 35).

Of the respondents who talked about a characteristic they learned about themselves, 25.0% (n = 60) developed a sense of self-worth, 32.9% (n = 79) boosted their self-confidence, 44.2%

**Table 4. Description of PHA experiences.**

| | Yes | | No | | Testing for significant difference between Yes and No $\chi^2$ (df = 1) | p-Value |
|---|---|---|---|---|---|---|
| | Percent | n | Percent | n | | |
| **What did your experiences as a PHA teach you about yourself this quarter?** | | | | | | |
| New Skills/Attitudes | 71.7% | 172 | 28.3% | 68 | - | - |
| Communication Skills | 49.2% | 118 | 50.8% | 122 | 0.067 | 0.7963 |
| Conflict Management | 20.4% | 49 | 79.6% | 191 | 84.017 | <0.0001* |
| Time Management | 7.5% | 18 | 92.5% | 222 | 173.40 | <0.0001* |
| Open-Minded | 14.6% | 35 | 85.4% | 205 | 120.42 | <0.0001* |
| Strengths/Limitations | 73.7% | 175 | | | | |
| Self-Worth | 25.0% | 60 | 75.0% | 180 | 60.00 | <0.0001* |
| Self-Confidence | 32.9% | 79 | 67.1% | 161 | 28.017 | <0.0001* |
| Self-Realization about Strengths/Limitations | 44.2% | 106 | 55.8% | 134 | 3.267 | 0.071 |
| Patience | 10.8% | 26 | 89.2% | 214 | 147.27 | <0.0001* |
| **What was your best/favorite experience of being an PHA this quarter?** | | | | | | |
| Building Connections | 66.8% | 163 | 33.2% | 81 | 27.55 | <0.0001* |
| Building Relationships with a Diverse Group | 22.1% | 54 | 77.9% | 190 | 75.80 | <0.0001* |
| Community Building | 24.6% | 60 | 75.4% | 184 | 63.02 | <0.001* |
| Helping Others | 41.4% | 101 | 58.6% | 143 | 7.23 | 0.0072 |
| **What did you learn about public health from being a PHA?** | | | | | | |
| Infectious Disease Prevention | 40.9% | 90 | 59.1% | 130 | 7.27 | 0.0070* |
| Multidisciplinary | 20.5% | 45 | 79.6% | 175 | 76.82 | <0.0001* |
| Community Effort | 36.8% | 81 | 63.2% | 139 | 15.29 | <0.0001* |
| Challenging | 15.5% | 34 | 84.6% | 186 | 105.018 | <0.0001* |
| Essential | 21.4% | 47 | 78.6% | 173 | 72.16 | <0.0001* |

Note: Due to missing values, the number of responses differed for each question. Data from questions 1, 2 and 3.

(n = 106) learned about their strengths/limitations, and 10.8% (n = 26) reported finding they were more patient than they realized.

When asked, "What was your best/favorite experience of being an PHA this quarter?," 90.6% (n = 221) reported enjoying networking with peers, faculty and community. This included building: (1) connections (66.8%, n = 163), (2) relationships with a diverse group (22.1%, n = 54), (3) community (24.6%, n = 60), and (4) helping others (41.4%, n = 101).

When asked, "What did you learn about public health from being a PHA?," 40.9% (n = 90) discussed gaining an understanding of infectious disease prevention, 20.5% (n = 45) recognizing public health is a multi-disciplinary field, 36.8% (n = 81) and that public health involves a community effort. There were 15.5% (n = 34) who discussed the challenges of public health, and 21.4% (n = 47) learned that public health was essential.

## Discussion

In this paper, we aimed to answer two questions: (1) Using a practicum framework, what are important considerations in designing and building a public health workforce for a university campus? and (2) What are the benefits to the workforce in terms of public health education and growth? The practicum framework allowed the program to quickly build a public health workforce. CEPH emphases guided the interview questions focus on: (1) attitude toward COVID-19 guidelines, (2) communication skills, (3) teamwork, and (4) customer service; this helped the program to build on PHA interests and skills. The result was the hiring of 282 undergraduate students from over 56 diverse academic majors. It also ensured a fit between student interests and the PHA role. This was reflected in the fact that there was little attrition during the 10-week fall quarter and the majority of students continued into the winter quarter. Furthermore, only 5% of students reported not being satisfied with their position.

Additionally, none of the PHAs contracted COVID-19 from exposure in their role as a PHA. Considering college campuses were identified as exposing students to high risk of COVID-19, this highlights the safety of the program [16] and the effectiveness of the University's COVID-19 public health policies. Emphasis on physical distancing, masking, peer education, and community involvement allowed for the creation of a safe work environment [3].

Modeling our program on public health educational guidelines gave students hands-on experience while facilitating learning and fostering a sense of community. Community building was an integral part of our program, with students forging relationships not only with the community and the population they served, but also with their PHA peers. Our results are consistent with previously published reports suggesting that practicums supplement students' academic and professional lives by giving them opportunities to network and to develop a sense of responsibility [17]. In addition, students indicated community building was a protective factor, allowing them to build relationships and remain connected in a safe environment. A safe, supportive, and community-based work environment is of great benefit to workforces in terms of physical and mental well-being and sense of community [18].

Our results also indicate students gained key knowledge about public health as well as insights about themselves. They learned about the role of public health, infectious disease processes and prevention, and the multidisciplinary nature of public health and health programs [14].

The PHA role gave students a sense of purpose during these uncertain times that helped to protect them from the negative effects of stress. As PHAs, they felt useful and that they were engaged in meaningful activities while learning about public health work. Having purpose in life is an important factor to building resilience, which helps prevent mental health disturbances and ensuring expedient recovery from major adverse events [19].

The PHA role also encouraged and supported professional growth and development. For example, students reported repeatedly needing to employ communication skills, and time management skills to perform their public health duties. These skills will prepare them for their careers no matter which field they pursue. As they mastered these skills, they grew in self-confidence and discovered the importance of acceptance and patience.

## Limitations and challenges

The limitations of our PHAP experience should also be noted. This program drew on the expertise of its public health sciences program. Public Health Sciences graduate students were integral to the hiring and supervising processes, and their coursework and past work experience enabled them to effectively model effective public health education for the undergraduates. This approach may not be fully replicable for universities without a public health educational program. But, they may be able to find other related fields from which to draw.

A major challenge the program faced was working with student schedules. Class schedules change each academic term and within the term, there is need for flexibility around midterms, finals, and vacation breaks. In response, two full-time Health Educators were hired to provide consistency and to supervise PHAs during shifts. In addition, the necessary flexibility when coordinating student schedules also required a full-time scheduler.

## Conclusions

The public health practicum framework and CEPH guidelines outlined a structure that helped to develop a quick response to the request for a public health workforce to address COVID-19 on campus. It also allowed us to identify the pool of students who would be successful in accomplishing the PHAP mission. It also created an infrastructure to expeditiously educate and prepare a large workforce while providing ongoing education about new, salient, and rapidly changing scientific knowledge and public health policies to a potentially confused and frustrated populace.

In turn, our students learned about public health, its role, and community importance in public health implementation. The PHAP structure and support fostered a safe environment in which students were able to feel part of the larger community while gaining valuable work experience and skills.

## Acknowledgments

The authors would like to acknowledge the contributions of the Amber Carrere, Kristyn Keylon, Alexis Calinawan, Awais Khan, Sheila Tolentino, and Dr. Cory Vu to development and ongoing management of the program. We would also like to thank the students who served as public health ambassadors and helped the campus to safely get through the year.

## Author Contributions

**Conceptualization:** Carolyn S. Dewa, Andrea M. Guggenbickler, Rebecca Phan.

**Data curation:** Carolyn S. Dewa, Zoe Che, Rebecca Phan.

**Formal analysis:** Carolyn S. Dewa, Zoe Che, Andrea M. Guggenbickler, Rebecca Phan.

**Funding acquisition:** Bradley Pollock.

**Investigation:** Zoe Che.

**Methodology:** Carolyn S. Dewa, Zoe Che, Andrea M. Guggenbickler, Rebecca Phan.

**Project administration:** Carolyn S. Dewa, Zoe Che, Andrea M. Guggenbickler, Rebecca Phan.

**Resources:** Bradley Pollock.

**Supervision:** Carolyn S. Dewa.

**Writing – original draft:** Carolyn S. Dewa, Zoe Che, Andrea M. Guggenbickler, Rebecca Phan.

**Writing – review & editing:** Carolyn S. Dewa, Zoe Che, Andrea M. Guggenbickler, Rebecca Phan, Bradley Pollock.

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
