## [Decision Letter · Decision Letter 0]

7 Feb 2022

PONE-D-21-21695Building a public health workforce for a university campus during a pandemic using a practicum framework: design and outcomesPLOS ONE

Dear Dr. Dewa,

Thank you for submitting your manuscript to PLOS ONE. After careful consideration, we feel that it has merit but does not fully meet PLOS ONE’s publication criteria as it currently stands. Therefore, we invite you to submit a revised version of the manuscript that addresses the points raised during the review process.

We look forward to receiving your revised manuscript.

Kind regards,

Anand Nayyar, Ph.D.

Academic Editor

PLOS ONE

Journal Requirements:

2. Please note that in order to use the direct billing option the corresponding author must be affiliated with the chosen institute. Please either amend your manuscript to change the affiliation or corresponding author, or email us at plosone@plos.org with a request to remove this option.

4. We note you have included a table to which you do not refer in the text of your manuscript. Please ensure that you refer to Table 3 in your text; if accepted, production will need this reference to link the reader to the Table.

Reviewers' comments:

Reviewer's Responses to Questions

**Comments to the Author**

1. Is the manuscript technically sound, and do the data support the conclusions?

Reviewer #1: Yes

Reviewer #2: Yes

2. Has the statistical analysis been performed appropriately and rigorously? 

Reviewer #1: Yes

Reviewer #2: Yes

3. Have the authors made all data underlying the findings in their manuscript fully available?

Reviewer #1: Yes

Reviewer #2: Yes

4. Is the manuscript presented in an intelligible fashion and written in standard English?

Reviewer #1: Yes

Reviewer #2: Yes

5. Review Comments to the Author

Reviewer #1: This paper describes the use and outcomes of a practicum

framework to create a university-based public health workforce to limit the spread of

COVID-19 on a university campus. We address two questions: (1) Using a practicum

framework, what are important considerations in designing and building a public health

workforce for a university campus? and (2) What are the benefits to the workforce in

terms of public health education and professional growth?

Methods: As part of its COVID-19 reopening plan, a West Coast university developed

a public health workforce comprised of 282 undergraduates tasked with encouraging

compliance with COVID-19 mitigating healthy behaviors. Program secondary data

were used to describe who was included in the workforce and their learning outcomes.

The majority indicated they learned new skills/developed new attitudes (71.7%) and

became aware of strengths and opportunities for professional growth (73.7%). The

types of new skills and attitudes learned included communication (49.2%), conflict

management (20.4%), time management (7.5%), and open-mindedness/less

judgmental attitude (14.6%). What did they learn about public health? They gained an

understanding of infectious disease prevention (40.9%), that it is a multi-disciplinary

field (20.5%), and it involves a community effort (36.8%).

Comments to work upon:

1. Abstract need to be restructured

2. Introduction lacks contribution and structure of what the other section will provide.

3. Abstract should reflect the background knowledge on the problem addressed need to be added.

4. Abstract should reflect the wide range of applications and its possible solutions need to be added.

5. Abstract should reflect the problem addressed need to be justified with more details.

6. In Introduction section, the drawbacks of each conventional technique should be described clearly.

7. Introduction section can be extended to add the issues with respect to existing work

8. What is the motivation of the proposed work?

9. Literature review techniques have to be strengthened by including the issues in the current system and how the author proposes to overcome the same

10. Research gaps, objectives of the proposed work should be clearly justified.

11. The conclusion should state scope for future work.

12. Authors should include some graphs, flowcharts for better presentation of work

Reviewer #2: 1. Are there any patterns across academic majors in terms of STEM vs. non-STEM students?

2. Are there statistically significant differences in perceptions due to their gender?

3. Additional factors that the authors could have considered include compensation, assignment of trivial work, competitive co-interns, and work overload.

6. PLOS authors have the option to publish the peer review history of their article (what does this mean?). If published, this will include your full peer review and any attached files.

Reviewer #1: No

Reviewer #2: No

---

## [Author Response · Author response to Decision Letter 0]

13 Apr 2022

REVIEWER COMMENTS IN CAPS.

1. ABSTRACT NEED TO BE RESTRUCTURED.

Thank you for pointing this out. As suggested, the Abstract has been restructured.

2. INTRODUCTION LACKS CONTRIBUTION AND STRUCTURE OF WHAT THE OTHER SECTION WILL PROVIDE.

Text in the Introduction was revised with clarification,

Using a practicum framework, a West Coast university with an undergraduate enrollment of 31,162 students, developed a public health workforce of undergraduate peer educators to address its reopening plans and meet its COVID-19 health promotion mandate. In this paper, we describe the use and outcomes of a practicum framework to create a university-based public health workforce to limit the spread of COVID-19 on a university campus. We address two questions: (1) Using a practicum framework, what are important considerations in designing and building a workforce for a university campus? and (2) What are the benefits to the workforce in terms of public health education and professional growth? This paper contributes to filling the gap in the literature about how a practicum framework can be used on a large scale as an effective learning tool in an urgent public health crisis.

3. ABSTRACT SHOULD REFLECT THE BACKGROUND KNOWLEDGE ON THE PROBLEM ADDRESSED NEED TO BE ADDED. 

As suggested, a Background section was added. It reads:

Background: The reopening of college and university campuses was seen as presenting a high risk for transmission of COVID-19. Thus, these institutions faced with a new public health challenge never heretofore faced on this scale. To magnify the problem, they needed to rapidly develop and implement re-opening plans in an environment filled with uncertainty and for a population that was significantly less likely to observe COVID-19 mitigation behaviors. In response, within three weeks of opening, as part of its COVID-19 public health strategy, a West Coast university created and trained a public health workforce comprised of 282 undergraduates tasked with encouraging compliance with COVID-19 mitigating healthy behaviors.

4. ABSTRACT SHOULD REFLECT THE WIDE RANGE OF APPLICATIONS AND ITS POSSIBLE SOLUTIONS NEED TO BE ADDED. 

As suggested, text has been added:

These findings demonstrate an effective way of rapidly addressing public health concerns that allowed for on the job training and opportunities for young adults to learn and grow.

5. ABSTRACT SHOULD REFLECT THE PROBLEM ADDRESSED NEED TO BE JUSTIFIED WITH MORE DETAILS.

As suggested, additional text 

Background: The reopening of college and university campuses was seen as presenting a high risk for transmission of COVID-19. Thus, these institutions faced with a new public health challenge never heretofore faced on this scale. To magnify the problem, they needed to rapidly develop and implement re-opening plans in an environment filled with uncertainty and for a population that was significantly less likely to observe COVID-19 mitigation behaviors. In response, within three weeks of opening, as part of its COVID-19 public health strategy, a West Coast university created and trained a public health workforce comprised of 282 undergraduates tasked with encouraging compliance with COVID-19 mitigating healthy behaviors.

6. IN INTRODUCTION SECTION, THE DRAWBACKS OF EACH CONVENTIONAL TECHNIQUE SHOULD BE DESCRIBED CLEARLY.

As suggested, text was revised and added:

But, to provide effective education, peers must be prepared with knowledge and communication skills. In public health, practicum structures are used as a teaching vehicle. Practicums present students opportunities to apply public health knowledge and skills in practice [12]. Undergraduate practicum projects have been shown to be highly effective to train students for public health work [13]. Thus, the practicum is considered a capstone in an undergraduate student’s public health experience [14]. This assumes that the practicum is being offered within the context of a public health major, raising the question of the effectiveness of a practicum framework to introduce and teach public health to students who are not majoring in public health. Furthermore, there is a paucity of literature on the use of the practicum model on a large scale to address a community’s public health needs that a pandemic demands. These demands include continuously keeping up to date with quickly changing scientific knowledge, public health policies, and educating a potentially frustrated public while creating a positive learning experience.

7. INTRODUCTION SECTION CAN BE EXTENDED TO ADD THE ISSUES WITH RESPECT TO EXISTING WORK.

As suggested, text was revised and added:

But, to provide effective education, peers must be prepared with knowledge and communication skills. In public health, practicum structures are used as a teaching vehicle. Practicums present students opportunities to apply public health knowledge and skills in practice [12]. Undergraduate practicum projects have been shown to be highly effective to train students for public health work [13]. Thus, the practicum is considered a capstone in an undergraduate student’s public health experience [14]. This assumes that the practicum is being offered within the context of a public health major, raising the question of the effectiveness of a practicum framework to introduce and teach public health to students who are not majoring in public health. Furthermore, there is a paucity of literature on the use of the practicum model on a large scale to address a community’s public health needs that a pandemic demands. These demands include continuously keeping up to date with quickly changing scientific knowledge, public health policies, and educating a potentially frustrated public while creating a positive learning experience.

8. WHAT IS THE MOTIVATION OF THE PROPOSED WORK?

As suggested, text was added:

This paper contributes to filling the gap in the literature about how a practicum framework can be used on a large scale as an effective learning tool in an urgent public health crisis.

It also created an infrastructure to expeditiously educate and prepare a large workforce while providing ongoing education about new, salient, and rapidly changing scientific knowledge and public health policies to a potentially confused and frustrated populace.

9. LITERATURE REVIEW TECHNIQUES HAVE TO BE STRENGTHENED BY INCLUDING THE ISSUES IN THE CURRENT SYSTEM AND HOW THE AUTHOR PROPOSES TO OVERCOME THE SAME.

The explanation was added:

But, to provide effective education, peers must be prepared with knowledge and communication skills. In public health, practicum structures are used as a teaching vehicle. Practicums present students opportunities to apply public health knowledge and skills in practice [12]. Undergraduate practicum projects have been shown to be highly effective to train students for public health work [13]. Thus, the practicum is considered a capstone in an undergraduate student’s public health experience [14]. This assumes that the practicum is being offered within the context of a public health major, raising the question of the effectiveness of a practicum framework to introduce and teach public health to students who are not majoring in public health. Furthermore, there is a paucity of literature on the use of the practicum model on a large scale to address a community’s public health needs that a pandemic demands. These demands include continuously keeping up to date with quickly changing scientific knowledge, public health policies, and educating a potentially frustrated public while creating a positive learning experience.

This paper contributes to filling the gap in the literature about how a practicum framework can be used on a large scale as an effective learning tool in an urgent public health crisis.

10. RESEARCH GAPS, OBJECTIVES OF THE PROPOSED WORK SHOULD BE CLEARLY JUSTIFIED.

The explanation was added:

But, to provide effective education, peers must be prepared with knowledge and communication skills. In public health, practicum structures are used as a teaching vehicle. Practicums present students opportunities to apply public health knowledge and skills in practice [12]. Undergraduate practicum projects have been shown to be highly effective to train students for public health work [13]. Thus, the practicum is considered a capstone in an undergraduate student’s public health experience [14]. This assumes that the practicum is being offered within the context of a public health major, raising the question of the effectiveness of a practicum framework to introduce and teach public health to students who are not majoring in public health. Furthermore, there is a paucity of literature on the use of the practicum model on a large scale to address a community’s public health needs that a pandemic demands. These demands include continuously keeping up to date with quickly changing scientific knowledge, public health policies, and educating a potentially frustrated public while creating a positive learning experience.

This paper contributes to filling the gap in the literature about how a practicum framework can be used on a large scale as an effective learning tool in an urgent public health crisis.

11. THE CONCLUSION SHOULD STATE SCOPE FOR FUTURE WORK.

As suggested, text was added to suggest further work, 

This approach may not be fully replicable for universities without a public health educational program. But, they may be able to find other related fields from which to draw. Future work could pursue innovative ways of replicating a similar workforce through academic and governmental partnerships.

12. Authors should include some graphs, flowcharts for better presentation of work

As suggested, the flowchart was moved from supplemental material to the manuscript.

Reviewer #2: 

1. ARE THERE ANY PATTERNS ACROSS ACADEMIC MAJORS IN TERMS OF STEM VS NON-STEM STUDENTS?

We examine whether the differences in satisfaction by STEM. There were none. We included these analyses in Table 3 and the text,

Of the PHA respondents, 98.8% rated their satisfaction with the PHA role (Table 3). The majority indicated they were satisfied with their role (95.2%, n = 238). Few were dissatisfied with their role either (4.8%, n = 12. In addition, 96.8% (n = 242) were satisfied the educational value of the program and 94.0% (n. =235) were satisfied with the value for their professional interests. There were no significant differences by either gender or STEM/Non-STEM major. Furthermore, none of the PHAs developed COVID-19 as a result of their role.

2. ARE THERE STATISTICALLY SIGNIFICANT DIFFERENCES IN PERCEPTIONS DUE TO THEIR GENDER? 

We examine whether there were differences in satisfaction by gender. There were none. We included these analysis in Table 3 and the text,

Of the PHA respondents, 98.8% rated their satisfaction with the PHA role (Table 3). The majority indicated they were satisfied with their role (95.2%, n = 238). Few were dissatisfied with their role either (4.8%, n = 12. In addition, 96.8% (n = 242) were satisfied the educational value of the program and 94.0% (n. =235) were satisfied with the value for their professional interests. There were no significant differences by either gender or STEM/Non-STEM major. Furthermore, none of the PHAs developed COVID-19 as a result of their role.

3. ADDITIONAL FACTORS THAT THE AUTHORS COULD HAVE CONSIDERED COULD HAVE CONSIDERED INCLUDE COMPENSATION, ASSIGNMENT OF TRIVIAL WORK, COMPETITIVE CO-INTERNS, AND WORK OVERLOAD. 

All were paid the same rate. The APHAs were assigned the same roles. We were careful to ensure that their roles involved public health policy education and the role did not deviate from this function. Most worked 10 hours/week.

---

## [Decision Letter · Decision Letter 1]

13 Jun 2022

Building a public health workforce for a university campus during a pandemic using a practicum framework: design and outcomes

PONE-D-21-21695R1

Dear Dr. Dewa,

We’re pleased to inform you that your manuscript has been judged scientifically suitable for publication and will be formally accepted for publication once it meets all outstanding technical requirements.

Kind regards,

Anand Nayyar, Ph.D.

Academic Editor

PLOS ONE

Additional Editor Comments (optional):

The Paper stands Accepted.

Reviewers' comments:

Reviewer's Responses to Questions

**Comments to the Author**

1. If the authors have adequately addressed your comments raised in a previous round of review and you feel that this manuscript is now acceptable for publication, you may indicate that here to bypass the “Comments to the Author” section, enter your conflict of interest statement in the “Confidential to Editor” section, and submit your "Accept" recommendation.

Reviewer #1: All comments have been addressed

2. Is the manuscript technically sound, and do the data support the conclusions?

Reviewer #1: Yes

3. Has the statistical analysis been performed appropriately and rigorously? 

Reviewer #1: Yes

4. Have the authors made all data underlying the findings in their manuscript fully available?

Reviewer #1: Yes

5. Is the manuscript presented in an intelligible fashion and written in standard English?

Reviewer #1: Yes

6. Review Comments to the Author

Reviewer #1: Paper title: Building a public health workforce for a university campus during a pandemic using a practicum framework: design and outcomes

Discusses Well: The majority of students indicated that through the practicum, they learned

new skills/developed new attitudes (71.7%) and became aware of their own strengths

and opportunities for professional growth (73.7%). The types of new skills and attitudes

learned included communication (49.2%), conflict management (20.4%), time

management (7.5%), and open-mindedness/less judgmental attitude (14.6%). In terms

of public health, they gained an understanding of infectious disease prevention (40.9%)

that is multi-disciplinary (20.5%), and involves a community effort (36.8%).

Conclusions: These findings demonstrate an effective way of rapidly addressing

public health concerns that allowed for on the job training and opportunities for young

adults to learn and grow. The practicum framework allowed the expeditious

development of a public health workforce that ensured a fit between student interests

and the role. This led to high retention with the majority of students continuing into the

winter quarter. Only 5% of students reported not being satisfied with their position.

None of the students contracted job-related COVID-19.

The role gave students a sense of purpose during the pandemic’s uncertain times that

helped to protect them from the negative effects of stress. The practicum structure and

support fostered a safe environment in which students were able to feel part of the

larger community while gaining valuable work experience and skills and serve their

community

As a conclusion, the technical content is good. Therefore, the contribution of this article is also satisfactory. I am accepting article for publication in this journal.

7. PLOS authors have the option to publish the peer review history of their article (what does this mean?). If published, this will include your full peer review and any attached files.

Reviewer #1: No

---

## [Editor Report · Acceptance letter]

12 Jul 2022

PONE-D-21-21695R1 

Building a public health workforce for a university campus during a pandemic using a practicum framework: design and outcomes 

Dear Dr. Dewa:

I'm pleased to inform you that your manuscript has been deemed suitable for publication in PLOS ONE. Congratulations! Your manuscript is now with our production department. 

Kind regards, 

on behalf of

Dr. Anand Nayyar 

Academic Editor

PLOS ONE